# Genome-resolved correlation mapping links microbial community structure to metabolic interactions driving methane production from wastewater

Brandon Kieft [1], Niko Finke[1], Ryan J. McLaughlin[1,2], Aditi N. Nallan[1,2], Martin Krzywinski[3], Sean A. Crowe [1,4] & Steven J. Hallam [1,2,5,6,7,8] ✉

Anaerobic digestion of municipal mixed sludge produces methane that can be converted into renewable natural gas. To improve economics of this microbial mediated process, metabolic interactions catalyzing biomass conversion to energy need to be identified. Here, we present a two-year time series associating microbial metabolism and physicochemistry in a full-scale wastewater treatment plant. By creating a co-occurrence network with thousands of time-resolved microbial populations from over 100 samples spanning four operating configurations, known and novel microbial consortia with potential to drive methane production were identified. Interactions between these populations were further resolved in relation to specific process configurations by mapping metagenome assembled genomes and cognate gene expression data onto the network. Prominent interactions included transcriptionally active *Methanolinea* methanogens and syntrophic benzoate oxidizing *Syntrophorhabdus*, as well as a Methanoregulaceae population and putative syntrophic acetate oxidizing bacteria affiliated with Bateroidetes (Tenuifilaceae) expressing the glycine cleavage bypass of the Wood–Ljungdahl pathway.

Renewable natural gas (RNG), primarily composed of biogenic methane ($CH_4$) and carbon dioxide is an important non-fossil energy resource useful in the transition to a low carbon future[1]. Despite widespread adoption of technologies such as anaerobic digestion (AD) to produce biogenic $CH_4$, industrial-scale AD converting organic waste (e.g., municipal black and gray water) to RNG tends to experience operational challenges, including (i) variable RNG yields, (ii) lower production efficiencies than theoretical values, and (iii) substantial amounts of solid residues that can be costly to dispose[2]. These challenges can confound the economics of RNG production and arise in part from a prevailing "black box" paradigm that does not fully consider the microbial communities, also known as microbiomes, driving AD conversion processes[3]. Discovering design principles that shape the network properties of AD microbiomes offers a new paradigm for optimizing RNG production and waste resource recovery.

Over the past decade, high-throughput sequencing approaches have been used to describe microbial community structure, function and dynamics associated with AD at different operating scales[4–9],

[1]Department of Microbiology and Immunology, University of British Columbia, Vancouver, BC V6T 1Z1, Canada. [2]Graduate Program in Bioinformatics, University of British Columbia, Vancouver, BC V6T 1Z4, Canada. [3]Genome Sciences Centre, BC Cancer Agency, Vancouver, BC V5Z 4S6, Canada. [4]Department of Earth, Ocean, and Atmospheric Sciences, University of British Columbia, Vancouver, BC V6T 1Z1, Canada. [5]Genome Science and Technology Program, University of British Columbia, 2329 West Mall, Vancouver, BC V6T 1Z4, Canada. [6]Bradshaw Research Institute for Minerals and Mining (BRIMM), University of British Columbia, Vancouver, BC V6T1Z4, Canada. [7]Life Sciences Institute, University of British Columbia, Vancouver, BC V6T 1Z3, Canada. [8]ECOSCOPE Training Program, University of British Columbia, Vancouver, BC V6T 1Z3, Canada. ✉e-mail: shallam@mail.ubc.ca

including recent efforts to characterize the global microbiome of wastewater ADs[10]. Importantly, these studies indicate that the AD milieu supports regionally distinct microbial communities containing heterogeneous constituents engaged in conserved metabolic interactions driving organic waste conversion to $CH_4$[11]. In addition to microbial dynamics, considerable evidence also indicates that industrial-scale AD performance is strongly influenced by a variety of process parameters including substrate chemistry, loading rates, retention time, temperature, and metal micronutrient (e.g., Fe, Ni, Mo, Mn) concentration[12–15]. Taken together, an emerging scientific consensus[16–18] suggests that a new paradigm for understanding and improving AD systems should involve constructing microbial interaction networks in relation to physical and chemical parameter information under time-resolved conditions at relevant operating scales. This is particularly needed for industrial-scale operations which are difficult to access and study over unit time[4,8,18–25].

Here, we present a two-year time series study of the Lulu Island municipal wastewater treatment plant (WWTP) operated by Metro Vancouver in Richmond, British Columbia, Canada. Our study is intended to facilitate further understanding of microbial community structure, function, and dynamics in relation to industrial-scale RNG production. The Lulu Island WWTP uses standard practices and produces >5000 m³ of RNG per day from mixed sludge as a starting material. Replicated mixed sludge samples were collected biweekly from Lulu ADs and archived for DNA and RNA extraction. Small subunit ribosomal RNA (SSU or 16S rRNA) gene amplicons were generated from 116 replicated samples, including two periods of standard AD operation, a period of AD operation with an additional allochthonous waste stream, and a period that used serial (instead of parallel) AD operation. The resulting amplicon sequence variant (ASV) data was used to identify a core set of microorganisms across the time series, construct a co-occurrence network to generate statistically informed hypotheses related to potential syntrophic interactions, and, in combination with process parameter information, identify indicator microorganisms associated with different process configurations and relevant WWTP conditions such as final nitrate levels, volatile solids destroyed, total RNG produced, and RNG methane content. A subset of time series samples representing each process configuration was also selected for metagenomic whole genome shotgun and metatranscriptomic sequencing to produce metagenome-assembled genomes (MAGs) and estimate expression levels based on transcript read mapping. MAGs were associated with cognate ASV nodes in the co-occurrence network. Ultimately, these multi-omic datasets and statistical approaches offered potential mechanistic explanations (through encoded and expressed functions) for metabolic interactions between co-occurring microorganisms in the AD, providing insight into the metabolic network driving RNG production in the Lulu Island WWTP.

## Results and discussion

### Lulu Island waste resource recovery ecosystem

The Lulu Island WWTP operated by Metro Vancouver in Richmond, British Columbia, Canada (Longitude: −123.14498° or 123° 8′ 42″ W, Latitude: 49.11491° or 49° 6′ 54″ N) provides primary and secondary treatment of >30 billion liters of mixed-sourced wastewater from ~200,000 residents each year. Primary treatment includes a series of tanks where wastewater undergoes screening, aeration, mechanical separation, settling, and clarification. This effluent then enters a secondary treatment stream, where it is pumped through a trickling filter, solids-contact tank, secondary clarifier, and disinfection tank before being released into receiving water. A portion of sludge from the primary and secondary treatment process streams is mixed and thickened, then split equally into two mesophilic (38 °C) anaerobic digesters (ADs) manifesting a 30-day retention time. The ADs are typically operated in parallel with identical mixed waste inputs. Methane and other gases from the ADs are scrubbed to renewable

natural gas (RNG) which is either used to generate heat for Lulu Island WWTP operations or sold to FortisBC, the local distributor of natural gas.

Metro Vancouver provided physicochemical parameters and triplicate waste secondary sludge (WSS) samples from solid contact tanks and overflow digestate samples from anaerobic digesters (AD1 and AD2) on a biweekly basis between October 13, 2016 and December 12, 2018. During this time interval, ADs experienced four different process configurations (Standard Operation I, standard operation with chemically enhanced primary treatment sludge (CEPT Operation), Standard Operation II, and Serial Operation; Fig. 1A) which differed both in upstream primary and secondary wastewater treatment steps and in the flow of material into the ADs. Each configuration represented experimental perturbations of municipal operations to determine how different process parameters contribute to RNG production,

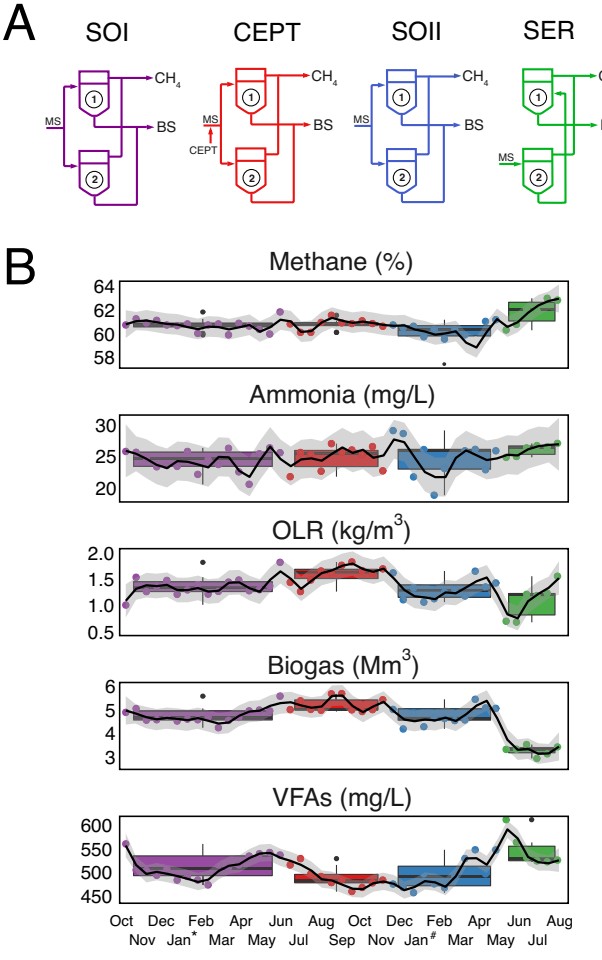

**Fig. 1 | Summaries of the operating conditions and physicochemical conditions of the digester across the time series. A** Wire diagram of the four process configurations across the two-year time series. SOI Standard Operation I, CEPT Chemically Enhanced Primary Treatment Operation, SOII Standard Operation II, SER Serial Operation, MS mixed sludge waste from primary and secondary treatment), $CH_4$ (biogas), BS biosolids, Labels "1" and "2" indicate the two anaerobic digesters (all samples for this study were taken from AD1). **B** Temporal physicochemical sparklines of AD performance. Methane % volumetric percent of biogas as methane, HRT hydraulic retention time, OLR organic loading rate, Biogas (total volumetric biogas generated), VFAs volatile fatty acids. Box plots of configuration groups represent 75th, 50th (median), and 25th percentiles, with whiskers representing 90th percentiles and outlier position as filled points above whiskers ($n = 16$ for SOI, 10 for CEPT, 11 for SO II, and 6 for SER).

volatile solids conversion, influent, and effluent sludge characteristics (e.g., nutrient and metal composition, chemical oxygen demand) in relation to microbial community structure, function, and dynamics. This study is focused on data collected and analyzed from one of the ADs (AD1), which was sampled continuously throughout the two-year time series.

## Physicochemical parameter information

Physicochemical parameter information was collected throughout the time series to evaluate both mixed sludge characteristics entering the AD as well as organic waste conversion processes e.g., solids destruction, VFA concentrations, RNG production (Supplementary Data 1). In pairwise comparisons, physicochemical profiles from the two Standard Operations (I and II) were indistinguishable (Supplementary Data 4). However, samples from the intervening CEPT Operation differed significantly from the two Standard configuration modes based on a sharp increase in organic loading (kg/m$^3$), which ultimately led to a temporary increase in total biogas production approaching 5739 m$^3$ (Fig. 1B). The final Serial Operation was the most distinct configuration of the four because of a fundamentally different flow regime, initiated by a rapid shift to in-line (serial) rather than equivalent flow of mixed sludge split between the two ADs (Fig. 1A). This caused the inflow rate (ML/day) into AD1 to increase sharply at the beginning of Serial Operation, along with a concomitant decrease in hydraulic retention time (days). During this interval, digestate characteristics such as volatile fatty acid concentration (mg/L), final ammonia in the effluent (mg/L), and percent CH$_4$ in RNG all increased. While the CH$_4$ to carbon dioxide (CO$_2$) ratio of RNG increased during the Serial configuration (indicating higher quality), the total volume of RNG decreased, suggesting a possible tradeoff in RNG quantity vs. quality which was then explored further from a microbiological perspective.

## Microbial community structure

Given the observed impact of process configuration on Lulu Island AD RNG production, we explored the sample archive to identify relationships between physicochemical parameters, such as VFA concentrations and RNG production, and microbial community structure. Genomic DNA extracted from 43 replicated AD1 samples spanning the time series was used to generate amplicons targeting the V4 region of the bacterial and archaeal (prokaryotic) 16S rRNA gene. Resulting data sets had an average of 23,833 quality-filtered 250-bp paired-end reads per sample resolving 928 unique ASVs (Supplementary Data 5). Consistent with physicochemical parameter information, microbial community structure also differed across samples in relation to process configuration (Supplementary Data 6). Interestingly, while the conditions in Standard Operation II returned to the same state as in Standard Operation I (after the intervening CEPT Operation), the microbial community did not return to its previous structure, indicating formation of a new stable state. The majority of unique ASVs (92%) had >80% sequence identity to cognate sequences in the global Microbes in Wastewater Treatment Systems and Anaerobic Digesters (MiDAS) 16S rRNA gene database[10], indicating that a phylogenetically and globally conserved set of microbial lineages are adapted to driving hydrolysis, fermentation and methanogenesis in the AD milieu (Fig. 2).

Despite high recall of ASVs to the MiDAS database, the structure of the Lulu Island AD community differed in several important ways from previously described datasets from mesophilic ADs (Fig. S1)[23,26–28]. Although Bacteroidetes, Firmicutes, and Proteobacteria were common and conserved community members, the most abundant taxonomic group in Lulu Island samples was Cloacimonetes (candidate phylum WWE1), with a cumulative relative abundance exceeding 25% in most samples across the time-series (Figs. S1, S2). This phylum is typically found in low abundance in mesophilic ADs treating wastewater, although it has been observed to dominate bioenergy facilities processing crop residues, and some mesophilic WWTPs (typically with

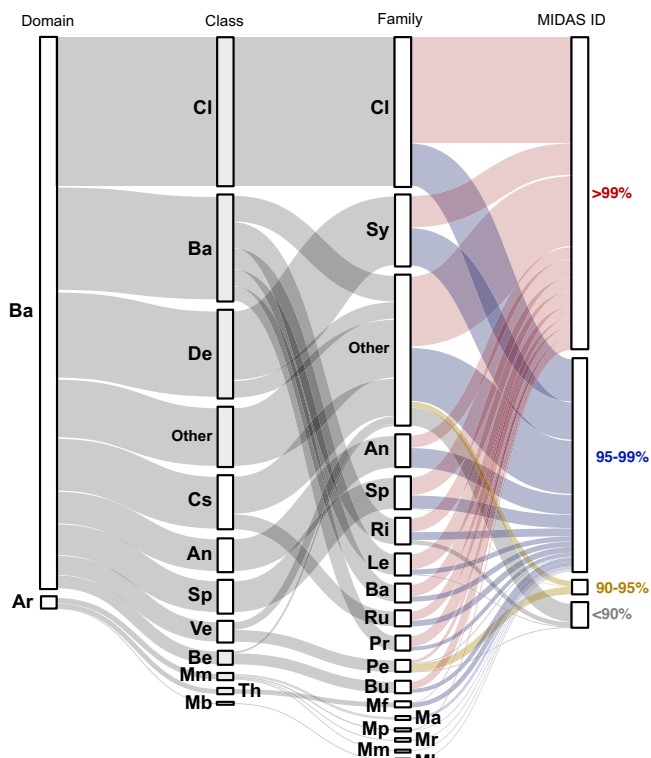

**Fig. 2 | A Sankey diagram of microbial community structure shown as relative abundances of family-level taxa grouped into their respective domains and classes.** The width of ribbons represents the cumulative relative abundances of all ASVs within each taxonomic lineage. The final column assigns ASVs to categories of percent identity of their V4 16S sequences to the MiDAS database. The two-letter codes on the plot represent taxonomic names. *Domain*: Ba Bacteria, Ar Archaea. *Class*: Cl Cloacimonadia, Ba Bacteroidia, De Deltaproteobacteria, Ot Other, Cs Clostridia, An Anaerolineae, Sp Spirochaetia, Ve Verrucomicrobiae, Be Betaproteobacteria, Mm Methanomicrobia, Th Thermococci, Mb Methanobacteria, *Family*: Cl Cloacimonadaceae, Sy Syntrophaceae, Ot Other, An Anaerolineaceae, Sp Spirochaetaceae, Ri Rikenellaceae, Le Lentimicrobiaceae, Ba Bacteroidetes (vadinHA17), Ru Ruminococcaceae, Pr Prolixibacteraceae, Pe Pedosphaeraceae, Bu Burkholderiaceae, Mf Methanofastidiosaceae, Ma Methanosaetaceae, Mp Methanospirillaceae, Mr Methanoregulaceae, Mm Methanomicrobiaceae, Mb Methanobacteriaceae.

long retention times or high organic loading rates), where it likely plays a role in amino acid fermentation and syntrophic propionate oxidation[8,29]. In addition to Cloacimonetes, ASVs associated with the candidate phylum Marinimicrobia were also relatively abundant, reaching up to 5.4% in some samples. Although prevalent and active in marine ecosystems under low oxygen conditions[30–32], Marinimicrobia are often observed in ADs where they may play a role in hydrogen production and nitrogen mineralization[33,34]. Additional candidate groups, including Atribacteria, Kiritimatiellaeota, and Hydrogenedentes, were also identified in Lulu Island samples at relative abundances approaching 1%, suggesting that these candidate groups are important AD community members whose metabolic roles are conserved across time[8,35].

Amplicon sequence variants affiliated with Methanogenic archaea had low relative abundances across the time series and included the families Methanofastidiosaceae (0.9%), Methanosaetaceae (0.5%), Methanospirillaceae (0.3%), Methanoregulaceae (0.2%), Methanomicrobiaceae (0.1%), and Methanobacteriaceae (0.1%). Methanofastidiosaceae, the most abundant methanogenic lineage in Lulu Island AD samples, is a candidate group that lacks several canonical methanogenesis pathway components, likely using methylated thiol compounds for energy and CH$_4$ production[33]. Methanosaetaceae, primarily

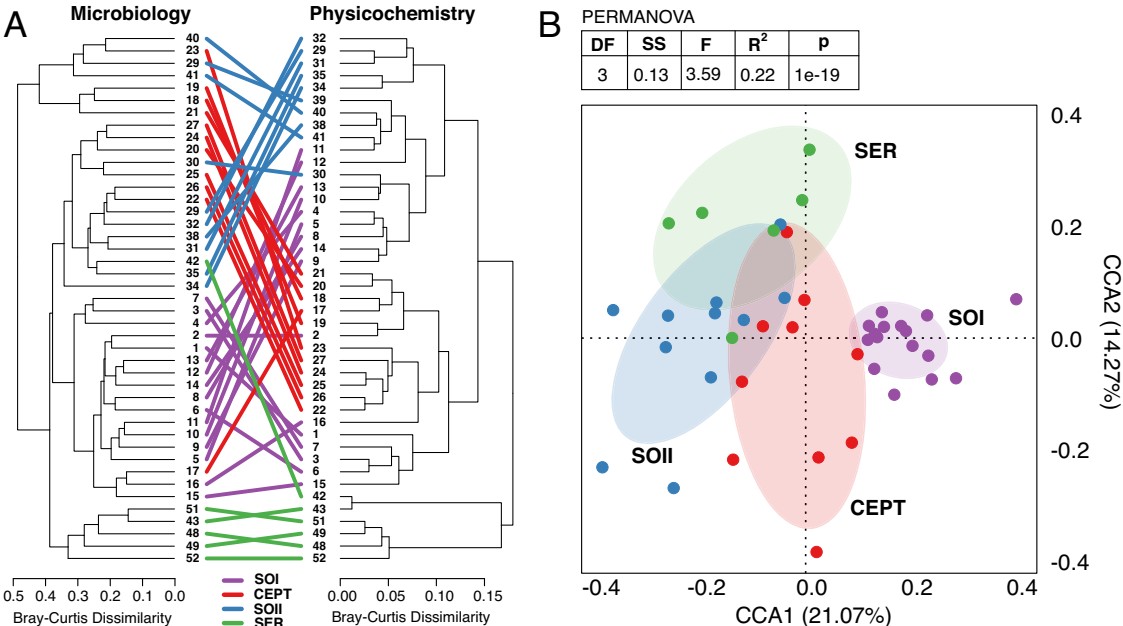

**Fig. 3 | Relationships between microbial community structure and physicochemical conditions. A** The structures of the AD microbial community (ASV data) and the AD physicochemical data are expressed as Bray-Curtis dissimilarity plotted in a dendrogram. Identical samples in each data-type dendrogram are connected with a line and colored corresponding to their process configuration. **B** The combined table of ASV and physicochemical data underwent principal coordinate analysis and samples were plotted in the first two coordinate dimensions. The variance explained by each dimension is shown as a percent in the axis text. Samples are colored by their process configuration and ellipses sized by the 95% confidence of the centroid of each configuration group are laid in the background. The two-sided PERMANOVA table above the plot shows results of a test between the four configuration groups.

represented by *Methanothrix* populations in Lulu Island samples, convert acetate to $CH_4$ without the need for syntrophic interactions, while the four remaining lineages are obligate hydrogenotrophic methanogens[36–38] that depend on syntrophic interactions with bacteria for electron equivalents driving $CO_2$ reduction to $CH_4$[39].

## Relationships between AD microbial communities and physicochemical parameters

In ecosystems with well-controlled resource inputs such as ADs with constrained feedstock, organic loading rate, retention time, temperature, etc., environmental conditions and microbial community structure can fluctuate together in stable patterns[4,40,41]. In the Lulu Island AD milieu, intervals with similar process configurations typically selected for similar microbial communities based on hierarchical cluster analysis (Fig. 3A), while sampling timepoints with distinct physicochemical parameters selected for distinct communities. Notably, at transition points between process configurations, when physicochemical parameters rapidly changed, microbial community composition rapidly responded in a time period shorter than the two-week resolution of our time series sampling, wherein microbial communities at the initial timepoint of a new operating condition still resembled that of the previous configuration before a new a new stable state was reached by the next sampling point (Fig. 3A).

By applying a dimension-reduction approach (i.e., canonical correspondence analysis) to a sample-by-sample distance matrix of the combined physicochemical parameter and ASV abundance information, process configuration was identified as a statistically significant variable in shaping the AD microbes and physicochemistry based on permutational analysis of variance tests (Fig. 3B). Using these combined datasets, it was observed that samples taken directly after transition to a new process configuration were more similar to the previous configuration. Given that no significant decline in RNG production was associated with these transitions (Supplementary Data 1), it appeared that detectable shifts in microbial community structure did not necessarily degrade AD performance with respect to overall

RNG yield. From these combined data, it was further determined from model fitting that the set of physicochemical parameters that most influenced microbial community structure was organic loading rate ($kg/m^3$) and the concentrations of volatile acids, ammonia, and nitrate (mg/L) in the AD digestate, accounting for 27.6% of the temporal variation in microbial structure observed across the time series (Supplementary Data 7). These key parameters are typically identified as the main drivers of microbial structure and activity in mesophilic WWTPs[42].

To account for population-level relationships between microbes and process configurations, ASV temporal distribution patterns were correlated with 28 measured physicochemical parameters across the time series. These results identified microbial populations and physicochemical parameters that were positively correlated (Fig. S4). One set of positively correlated variables included total biogas production ($m^3$), organic loading rate ($kg/m^3$), and several populations of Clostridia, Sphingobacteriales, Chloroflexi, and Spirochaetes. A second set of positive correlations, loosely associated with the first set, included AD hydraulic retention time (days), percent biogas composed of $CO_2$, concentration of effluent nitrate (mg/L), and several populations of Bacteroidales, Proteobacteria, and acetoclastic methanogens affiliated with Methanothrix. These groupings support previous observations that certain heterotrophic microorganisms, typically those engaged in hydrolysis coupled to fermentation, as well as acetoclastic methanogenic archaea, are most competitive when organic loading rate and retention time are high[43, 44]. Although total gas production was positively correlated with this set of conditions and taxa, the percentage of $CO_2$ was also high relative to other timepoints, indicating that a larger share of organic carbon input was metabolized to $CO_2$ rather than to $CH_4$. This correlation of higher $CO_2$ percent with dominance of acetoclastic methanogens is not unexpected given the stoichiometric imbalance in $CO_2:CH_4$ between different methanogenic pathways[45,46]. A third set of positively correlated variables included percent $CH_4$, temperature (°C), volatile fatty acids (mg/L), and several populations of Clostridiales, Bacteroidetes, and hydrogenotrophic methanogens

affiliated with Methanoculleus (Fig. S4). A fourth set, loosely correlated with the third, included BOD and COD, ammonia (mg/L) and suspended solids (mg/L), as well as hydrogenotrophic methanogens affiliated with Methanomicrobiales. Previous studies indicate that increases in both ammonia and fatty acid concentrations tend to select for hydrogenotrophic methanogenesis under mesophilic conditions[12,47–51]. Shifts from acetoclastic to hydrogenotrophic methanogenesis have also previously been observed in WWTP ADs[39,52,53], reducing the methane potential of effluent[54,55], and resulting in higher methane content of RNG[56].

Taken together, these results indicate significant coupling between physicochemical parameters and both population- and community-level microbial structure across process configurations. Methanogenic archaea and some putative syntrophic bacteria that cooperate to reduce $CO_2$ to $CH_4$ were strongly selected during the Serial Operation, while direct acetate-reducing methanogens and other fermentative and non-syntrophic bacteria were selected in other process configurations. Physicochemical parameter measurements generally corroborated these patterns, including higher ammonia associated with increased hydrogenotroph/syntroph abundances, higher VFA concentrations associated with increased abundances of fermentative bacterial lineages, and higher RNG $CO_2$ content associated with increased acetoclastic methanogen abundances. Subsequent investigation of the time series focused on identifying microbial indicators engaged in metabolic interactions with potential to influence RNG purity and yield.

### Identification of microbial indicators

During the time series, >25 complete volumetric turnover events occurred based on an average AD retention time of 30 days. Despite this recurring bottleneck, a robust core microbiome could be identified in the Lulu Island AD containing >30% of identified ASVs (339) in at least 80% of samples. This core collectively accounted for 64.4% of total 16S rRNA gene sequence reads. Given that input wastewater to the Lulu Island WWTP varied in origin and upstream processing across the time series, evidence of a robust core microbiome supports the hypothesis that selection factors such as environmental filtering and ecological interactions were likely more important than the initial community composition of waste inputs and the founder effect[11,57,58]. Based on previous work, selection pressures that help maintain a core AD microbiome include strong physical constraints such as retention time and organic loading[59], chemical constraints such as low trace metal concentrations and highly anoxic, reducing conditions[60], and biological constraints such as metabolic interactions between co-occurring microorganisms[61]. Research to identify and characterize these modes of selection in the AD milieu are becoming increasingly important to the biotechnology sector, typically with the assumption that operational controls can be identified and leveraged to select for communities that improve RNG production[6,20,28,62,63].

Although a robust core microbiome persisted throughout the time series, the relative abundance of many taxa was significantly impacted by shifts in physicochemical parameters and process configuration. Through indicator species analysis, with configuration as the conditional variable and all ASVs tested for significant over-representation based on temporal abundance patterns, 138 indicator ASVs were identified, including both common and conditionally rare taxa (Fig. S3). Indicator ASVs were usually distributed such that each configuration was represented by a unique set of ASVs, even within a single family-level taxon, suggesting there may be subtle underlying diversification patterns supporting functional redundancy in the AD milieu. For example, while there were many indicators affiliated with the Syntrophaceae family of Proteobacteria, there were unique sets of Syntrophaceae indicator ASVs for Standard I, CEPT, and Standard II, and Serial Operations. Other taxa with indicator ASVs were more specific in their representation of a given process configuration.

For example, 10/13 ASVs from the Rikenellaceae family of Bacteroidetes were affiliated with the Serial Operation. Another notable pattern was the partitioning of archaeal indicator ASVs between configurations based on methanogenic phenotype. For example, the sole archaeal indicator for Standard Operation I was a population of Methanosaetaceae, which perform acetoclastic methanogenesis, while the two archaeal indicators for Serial Operation were affiliated with Methanoregulaceae and Methanobacteriaceae, which only perform hydrogenotrophic methanogenesis. Taken together, indicator analysis at the ASV level provided further evidence that the activity of co-occurring populations, both as core and configuration-dependent consortia, helps shape the active community structure driving RNG production.

### Time-resolved correlation network analysis

Significant correlations were identified between AD process parameters, and both microbial community structure and the abundance of specific microbial populations relevant to RNG production over time. Based on the distributed nature of biomass conversion to methane between different microbial populations in WWTP ADs, we hypothesized that different process configurations would not only select for different populations but also that metabolic interactions among and between populations would vary under selection. To test this hypothesis, a co-occurrence network based on normalized ASV abundances across the two-year time series was constructed from a sparse inverse covariance matrix. Metadata information about nodes in the network, such as indicator ASV status, functional information from a paired MAG, and connectivity to other nodes, were then mapped onto the network to identify configuration-dependent subnetworks containing ensembles with potential to drive RNG production.

The ASV co-occurrence network was composed of 390 ASVs which had significant covariance across the time series (Fig. 4). The average clustering coefficient of the network (i.e., a measure of grouping or density among nodes) was 0.16, placing it well-within the range of microbial food webs or functional networks, and higher in connectivity than randomly produced associations between populations (Supplementary Data 2)[64]. ASVs which were indicators for a given configuration tended to be connected in the network (a result of covariance between the indicator analysis and the co-occurrence model) and formed strongly correlated subnetworks within the larger parent network. Similarly, closely related ASVs (e.g., those with >97% 16S V4 rRNA homology) tended to coalesce into subnetworks, suggesting that populations with low phylogenetic distance tended to share similar temporal distribution patterns and conserved functional roles due to similar selection pressures (see nodes and edges in Supplementary Data 8 and Supplementary Data 9).

### Genome-resolved correlation network mapping

Although taxonomically labeled ASVs can provide a way to predict trait-based information that is useful for inferring metabolic interactions, many AD microorganisms have poor taxonomic classifications and additional genome-resolved analysis is needed to assign potential functional roles[8,9,20,27,43]. Instead of using reference databases to infer functions from ASV nodes in the network, MAGs were generated from a set of 17 representative samples used for metagenomic whole genome shotgun sequencing across the time series with the goal of linking MAGs to ASVs and then ascribing functions to network nodes. The resulting metagenome datasets had an average assembled length of 657.4 Mbp, from which 40 high-quality MAGs (HQ: >90% complete and <5% contamination, with at least one full ribosomal RNA operon) and 475 medium-quality MAGs (MQ: >50% complete and <10% contamination, with at least one gene copy of the three ribosomal RNA subunits) were binned. Metagenome read mapping indicated that HQ and MQ bins represented on average 12.03% of the total quality base pairs sequenced per sample (Supplementary Data 2). The resulting

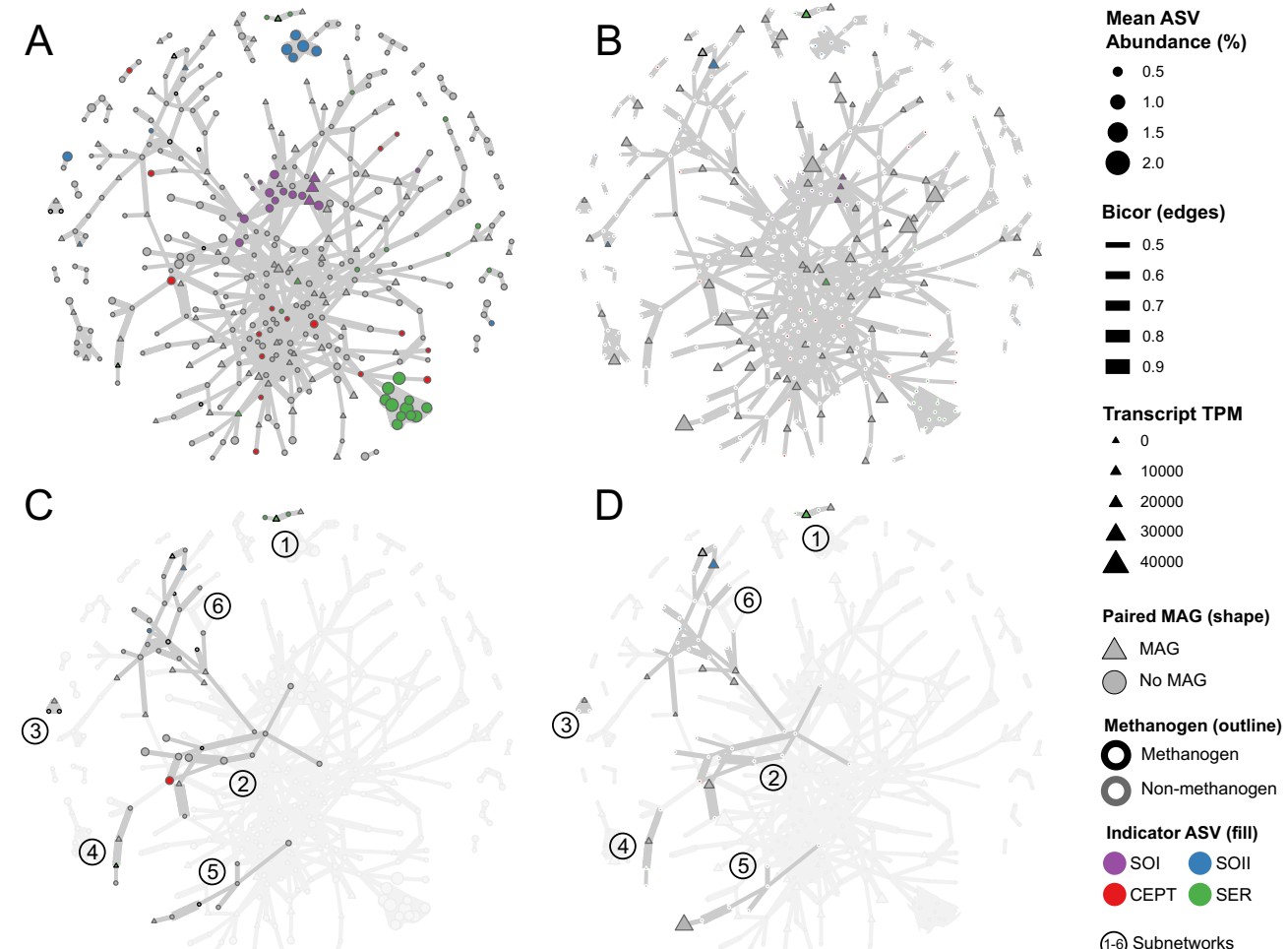

**Fig. 4 | A weighted co-occurrence network of the microbial community depicting multiple-test-corrected biweight midcorrelation values (edges) of ASVs (nodes) across the two-year time series.** Panels (**A**) and (**C**) depict the network wherein node sizes represent ASV mean relative abundance across all samples, while Panels (**B**) and (**D**) depict node sizes which represent transcript expression levels (TPM) of the representative node MAG. Nodes are shaded with color if they are indicator ASVs for a given process configuration, outlined in black if they are methanogen ASVs, and shaped as triangles if they have a representative MAG. Subnetworks are numbered and highlighted in Panels (**C**) and (**D**).

MAGs were mapped onto cognate ASVs based on 16S rRNA gene sequence homology, enabling genome-resolved analysis of nodes within the network. A total of 233 ASVs (25.1% of total ASVs) could be paired with a unique MAG at a 16S rRNA gene sequence homology cutoff of >99.5% over the full length of the V4 region (-1 allowed mismatch). Of the 233 ASV-MAG pairs identified by homology, a total of 186 shared the same lowest-common ancestor taxonomy between the ASV V4 region and the MAG genomic sequence and were used for genome-resolved correlation network mapping (see methods).

Thus, the ASV time-series data, enabled by high-resolution sampling, were used to build a network of co-occurring ASVs that characterized temporally coherent taxa which responded similarly to changes in WWTP process configuration. Then, metagenomic sequence information was used to overlay a functional architecture onto the network that used representative MAGs to describe potential metabolic interactions between co-occurring populations at the level of genes, reactions and pathways. The 186 ASV-MAG pairs were distributed throughout the parent network, suggesting good coverage of taxonomic lineages and process configurations (Fig. 4).

In addition to linking ASVs to MAGs via 16S rRNA gene identity, metatranscriptomes were generated from the 17 samples with metagenomes (Supplementary Data 3), allowing assignment of gene expression information (as transcripts per million, TPM) to MAGs in the network (Supplementary Data 10). A notable observation from this analysis was that methanogens displayed a strong decoupling between the metagenome- and metatranscriptomes-based measured relative abundances regardless of methanogenic lineage[65–67], most of which had low TPM in metagenomes but very high TPM in metatranscriptomes (Supplementary Data 10). However, when only considering bacterial MAGs and not methanogen MAGs, the correlation between metagenome TPM abundance and metatranscriptome TPM abundance was strong ($\rho = 0.53$; $t = 13.81$; df $= 472$; $p$-value $< 0.001$), indicating that bacterial relative abundance observed in metagenome libraries was predictive of enzymatic activity (metatranscriptomes) in the AD milieu.

## Subnetworks of methanogenic consortia

Taken together, the genome-resolved correlation network contained information about (1) temporal patterns of population abundance from the ASV time series, (2) indicator taxa for the various AD process configuration, (3) a subset of network nodes (47.7%) associated with MAGs, and (4) gene-level expression of each MAG. Six subnetworks of co-abundant methane-producing populations were identified within the parent network (Subnetworks 1–6), which were constructed by identifying hydrogenotrophic methanogen ASVs and all nodes connected by primary or secondary co-occurrence edges (one or two degrees of separation). These subnetworks represented ensembles with potential to drive RNG production across the time series, with

supporting evidence for biogas-producing syntrophic metabolisms gleaned from metabolic pathway reconstruction and gene expression data associated with MAGs mapped onto cognate ASVs within subnetworks.

Subnetwork 1 included four populations: three Rikenellaceae ASVs annotated as MiDAS species 240 (M.S.240) and M.S.3232, two of which were Serial Operation indicators (one of the Rikenellaceae indicator ASV had a corresponding MAG that was annotated to Tenuifilaceae in GTDB), and one Methanoregulaceae ASV (M.S.4938), which was also a Serial Operation indicator with an associated MAG (Supplementary Data 8). This small Subnetwork was isolated from the rest of the parent network and, given that 75% of nodes were indicator taxa, the organisms in this subnetwork were significantly more abundant during the Serial Operation when RNG purity was highest. The two MAGs in Subnetwork 1 were both high-quality, with completeness and contamination metrics of 93.3% and 1.4% (Tenuifilaceae; 3300028576_27) and 99.0% and 0% (Methanoregulaceae; 3300036947_39). Due to the strong co-occurrences during Serial Operation and the quality of the MAGs in Subnetwork 1, the encoded and expressed functions of the two genomes were examined for evidence of potential metabolic interactions. As expected, the Methanoregulaceae MAG encoded and expressed all genes necessary for hydrogenotrophic methanogenesis (e.g., *mcrABCDG*, genes Ga0377204_000018.861 – 865 and *mtrABCDEFH*, genes Ga0377204_000018.866 – 873) and lacked acetate kinase *(ack)* and phosphoacetyl transferase *(pta)* needed for acetoclastic methanogenesis (Figure S5; Supplementary Data 11). The Tenuifilaceae MAG that mapped to the subnetwork ASV encoded and expressed a partial Wood Ljungdahl (WL) pathway *(fdh, fhs, fol, met)* but lacked the CODH/ACS complex considered necessary for canonical acetate-oxidizing syntrophy by reverse WL[68] (Figure S5; Supplementary Data 11). The MAG did, however, both encode and express genes necessary for oxidizing acetate to 5,10-methylenetetrahydrofolate via the glycine cleavage system (Ga0255340_1000079.146, Ga0255340_1000129.100, and Ga0255340_1003402.5), indicating a potential route connecting acetate to its partial reverse WL methyl branch[33,56]. While this glycine cleavage system syntrophic mode has been proposed in several taxa there is currently no definitive evidence of its activity[33,56,62,69–71], and many organisms use this system in amino acid biosynthesis reactions and other diverse functions independent from the reverse WL pathway[72]. The Tenuifilaceae MAG also encoded and expressed organoheterotrophic functions common to other Bacteroidetes, including a sus-like biopolymer degradation and transport system and several peptidases (Supplementary Data 11), indicating that these three ASVs could also be co-occurring with Methanoregulaceae through non-syntrophic modes of interaction (e.g., by catalyzing the rate-limiting depolymerization steps upstream of methanogenesis or through co-selection because of similar preferences for the conditions in the AD under Serial Operation).

Subnetworks 2, 3, and 5, centered around *Methanobacterium lacus*, *Methanospirillium* M.S.2576, and *Methanospirillium* M.S.2576, respectively, none of which had associated high-quality MAGs. Subnetworks 2 and 5 each had an ASV from a syntrophic bacterial lineage (*Smithella* and Syntrophaceae) directly connected to the methanogen node (Supplementary Data 9), while Subnetwork 3 was an isolated group containing three nodes (two methanogen and one Paludibacteraceae ASVs). Subnetwork 2 was the only subnetwork containing hydrogenotrophic methanogens that also included a Cloacimonadaceae ASV despite the latter lineage exhibiting the highest total relative abundance in the time series analysis and representing 10.5% of nodes in the parent network. This suggests that Cloacimonadaceae are likely not closely associated with syntroph-dependent methanogens in the Lulu Island AD milieu.

Subnetwork 4 contained a Clostridia ASV (Christensenellaceae R-7 M.S.240) with an associated MAG (3300028677_44), a Bacteroidales

UCG-001 ASV (M.S.1138), a Synergistaceaae ASV (M.S.2022), and a *Methanobacterium lacus* ASV with an associated MAG (3300028677_53) that was also an indicator for Serial Operation (Supplementary Data 8). The two MAGs in Subnetwork 2 had completeness and contamination of 78.3% and 0.4% (Christensenellaceae) and 94.4% and 0.8% (Methanobacteriaceae), respectively (Supplementary Data 10), and the Christensenellaceae was directly connected with the syntrophy-dependent *Methanobacterium* methanogen. This Firmicutes family includes populations of peptide/amino acid fermenters and potentially $H_2$-producers[73,74], and though it has no isolated representatives, the Christensenellaceae R-7 lineage has been observed to comprise up to 6% of the AD community in some mesophilic digesters[10]. According to functional data from the Christensenellaceae MAG in Subnetwork 4, this population encoded and expressed several peptide and amino acid metabolism functions (Supplementary Data 11), including three major operons (two *livFGHKM* and one *azlCD*) for branched-chain amino acid transport, over 15 metabolic amino/exo/endo/oligopeptidase enzymes, and the hallmark fermentation enzyme pyruvate:ferredoxin oxidoreducase (Ga0255346_1000279.56). Between the Bacteroidales (as a depolymerizer[75]), Christensenellaceae (as a fermenter), Synergistaceae (as an SAOB[56,76]), and *Methanobacterium* (as a hydrogenotrophic methanogen) ASVs, Subnetwork 4 retained the metabolic capacity to carry out hydrolysis, fermentation, acetate oxidation, and methanogenesis of complex organic material to RNG. The observation that this *Methanobacterium* ASV was also a statistical indicator for the Serial Operation further suggests that this ensemble was most abundant in time series analysis during the period of high local RNG purity.

Subnetwork 6 was the largest co-occurring set of ASVs and included 25 nodes with 7 total ASV-MAG pairs (Supplementary Data 8). This Subnetwork was formed by four smaller but contiguous subnetworks. With four hydrogenotrophic methanogens in the 25-node subnetwork, Subnetwork 6 accounted for 40% of hydrogenotrophic methanogen ASVs in the parent network. Among the 25 nodes were also two indicator ASVs for Standard Operation II. Subnetwork 6 included several syntrophic lineages that had primary or secondary edges connected to the methanogen nodes, including two *Syntrophomonas* (both M.S.3971), one *Syntrophorhabdus* (M.S.998), and one Synergistaceae (*Thermovirga* M.S.988)[56,77]. Based on 10,000 permutations of 25 random nodes from the parent network, the probability of observing four syntroph nodes and four hydrogenotrophic methanogen nodes by chance was 2.8%, showing that Subnetwork 6 was rich in syntrophic interactions driving RNG production.

Nodes with representative MAGs in Subnetwork 6 were two Clostridia ASVs (D8A-2 lineage and *Ruminococcus*) and a single ASV from Desulfobacterota (*Syntrophorhabdus* M.S.998), Deltaproteobacteria (*Phaselicystis* M.S.2086a), Paludibacteraceae (M.S.2677), Patescribacteria (Candidatus Falkowbacteria M.S.5033), and *Methanolinea* (M.S.4938). The directly co-occurring *Methanolinea* and *Syntrophorhabdus* MAGs were 99.0% complete with 0% contamination (3300036947_39) and 72.9% complete with 8.6% contamination (3300028576_34), respectively, and each had a full rRNA operon. The 16S rRNA gene sequence of the *Syntrophorhabdus* MAG (across the whole 1482 bp gene) was 93.48% similar to the best cultured representative *Syntrophorhabdus aromaticivorans str. UI*. Similar to this type strain, the *Syntrophorhabdus* MAG encoded and expressed benzoate-CoA ligase for benzoate degradation to benzoyl-CoA (Ga0255340_1019139.2) and two benzoyl-CoA reductase operons for converting benzoyl-CoA to the dienoyl-CoA intermediate (Ga0255340_1011848.9 and Ga0255340_1011848.11, plus Ga0255340_1031544.4 and Ga0255340_1031544.6), including two operons of the heterodisulfide reductase involved in this endergonic step; the benzoyl-CoA reductase enzymes were highly expressed (TPM = 17.53; 30th most abundant ORF in the genome), indicating that this pathway is a critical metabolic step for *Syntrophorhabdus*

(Supplementary Data 11). To complete the oxidation of benzoate to $H_2$, $CO_2$, and fatty acids, the MAG encoded and expressed syntenic genes for dienoyl-CoA hydration (Ga0255340_1005473.7), breaking and hydrolyzing the intermediate ring (Ga0255340_1005473.3 and Ga0255340_1005473.5), and finally for beta-oxidation (Ga0255340_1013965.3-7).

Taken together, these observations provide an overlaid picture of population-level co-occurrence, metabolic interactions facilitating that co-occurrence, and gene expression data to support those metabolic interactions. This study also provides a model for conducting time series amplicon analysis coupled with genome-resolved correlation mapping to identify known and novel metabolic interactions underlying organic waste conversion to RNG with potential to define new design principles that improve RNG quality and yield at scale. These same design principles can in turn be extended to the production of other value-added compounds, hastening the transition from waste treatment to sustainable waste resource recovery.

## Methods

### Sample collection and processing

We conducted a time series sampling campaign of AD microbial community structure, function, and dynamics at the Lulu Island WWTP in order to identify taxa associated with enhanced RNG production. Metro Vancouver provided physicochemical process parameters and triplicate samples of overflow digestate from anaerobic digesters (AD1 and AD2) at the Lulu Island WWTP on a biweekly basis for two years resulting in a sample collection archive consisting of 28 different physicochemical parameter measurements as well as microbial DNA for amplicon and metagenomic whole genome shotgun sequencing, and RNA for metatranscriptomic sequencing across 43 sampling dates (Supplementary Data 1). After transport to the lab, mixed sludge samples were centrifuged for 15 min at $14,000 \times g$, followed by chemical flocculation of the remaining cells in the supernatant with $FeCl_3$ and $NaOH$[78]. After a final spin the supernatant was discarded, and the pellet stored at $-80\,°C$ prior to nucleic acid extraction. Mixed sludge DNA was extracted from frozen pellets using a DNeasy PowerSoil kit from Qiagen according to the manufacturer's specifications. DNA quantity and purity (260/280 values) were measured using a Nano-Drop (Thermo Fisher). Mixed sludge RNA was extracted from frozen pellets using RNeasy PowerSoil kit from Qiagen. RNA quantity and quality were measured using a NanoDrop.

### Small subunit ribosomal RNA gene amplicon sequencing

Microbial community structure and dynamics of mixed sludge samples was determined using PCR amplicon libraries targeting the V4 region of the bacterial and archaeal (prokaryotic) small subunit ribosomal RNA (SSU or 16S rRNA) gene using the 515F (GTGYCAGCMGCCGCGGTAA) and 806R (GGACTACNVGGGTWTCTAAT) primers according to Earth Microbiome Project guidelines[79]. A barcode was added to each amplicon library for multiplex sequencing on the Illumina MiSeq platform. The QIIME2 function demux was used to demultiplex the resulting FASTQ-formatted sequence files into separate samples, which were uploaded to the NCBI SRA database under PRJNA902729 as SRR22733982 – SRR22734080. The QIIME2 DADA2 pipeline was used to denoise amplicon libraries, filter chimeras, and create a feature table of amplicon sequence variants (ASVs)[80,81]. ASVs were filtered to remove those with less than one observation on average across all samples and assigned taxonomy based on the QIIME2 feature-classifier-sklearn function with confidence set to 0.75 using a local install of the Silva v132 database[82]. ASV sequences were also assigned taxonomy using the same method to the GTDB bacterial and archaeal 16S rRNA gene database (v202) to compare taxonomy to the paired MAGs (see later methods). 16S V4 rRNA gene amplicon sequences for all samples were uploaded to the NCBI GenBank database under BioProject PRJNA902729 as BioSamples SAMN32228586 - SAMN32228684.

### Amplicon sequence analysis and metadata integration

Physicochemical process parameters and metadata were integrated into the QIIME2 feature taxonomy and count table using the add-metadata and convert functions of biom to create a tab separated table of ASV designators, their taxonomic affiliation, their amplicon read count across samples, and metadata associated with each sample[81]. A batch correction was employed to remove any effects of sequencing date (sequence libraries were generated on three separate plates) as a factor explaining biological variation using the R packages DESeq2 and limma[83,84]. Some physicochemical parameters were not measured daily; therefore, an interpolation technique was used to estimate concurrent biological and physicochemical data. Briefly, any biological sampling dates missing from the physicochemical data table were added as an empty row, then interpolated using the smooth-fitting splinefun() function of the base R 'stats' package; parameters were checked manually by plotting to ensure reliable data interpolation between empirical physicochemical values. The 16S rRNA amplicon data (ASV phylogenetic tree, 16S rRNA gene sequences, ASV taxonomy, sample metadata) were imported into R and merged into a master phyloseq object using the function *merge_phyloseq()* of the 'phyloseq' R package[85].

### Statistical analyses and data visualization of ASV data

All analysis and plotting code for the ASV analysis in this work can be found in the Supplementary Software. Taxonomic community composition of ASVs (as an average across all samples) was plotted using the *sankeyNetwork()* function of the 'networkD3' R package. In the final category of the Sankey diagram, the percent identity of ASV V4 sequences to the MiDAS database was determined using a local install of the BLASTn software (v2.5.0) with default parameters[10,86].

Hierarchical clustering of both ASV and physicochemical data was performed using the *hclust()* function of the base 'stats' R package, with distance matrices calculated using the Bray-Curtis dissimilarity metric on the ASV count table and physicochemical metadata table exported from the master *phyloseq* object. The hierarchical clustering dendrograms from the two datasets were corresponded using the *tanglegram()* function of the 'dendextend' R package[87]. Canonical correspondence analysis (CCA) was performed using a Hellinger transformation of the combined physicochemical and ASV data with the *decostand()* and *cca()* functions of the 'vegan' R package[88]. The results were plotted using the 'ggplot2' R package[89]. Permutational analysis of variance (PERMANOVA) of samples across the four AD Configurations in the CCA was calculated with the *adonis()* function of the 'vegan' R package.

The community co-occurrence network was calculated by directly passing the master *phyloseq* object to the *spiec.easi()* function from the 'SpiecEasi' R package, with the glasso method, a minimum lambda ratio of 0.1, and the bstars selection criteria[90]. Nodes in the network represent ASVs observed in at least 25% of samples across the time series and are sized by mean relative abundance. Vertices in the network represent positive co-occurrence values, with vertex width being proportional to the co-occurrence value calculated from the *spiec.easi()* function.

Microbial taxonomic compositional data across individual samples and lineages was exported from the master *phyloseq* object and plotted in barplot and scatterplot form using functions within the 'ggplot2' R package. Indicator species analysis using the *multipatt()* function of the 'indicspecies' R package was used to derive statistically significant associations between ASVs and each configuration[91]. Each indicator ASV was grouped by taxon and results were plotted using 'ggplot2.

A maximum rank correlation test was performed to determine the set of physicochemical variables which best explained ASV community structure using the *bioenv()* function of the 'vegan' R package with default parameters. Correlations between temporal physicochemical parameter values and abundances of individual ASVs were calculated with the *cor()* function of the base 'stats' R package and plotted as a heatmap using 'ggplot2'.

## Metagenomic shotgun sequencing

A subset of 17 time series samples (Supplementary Data 1) were sequenced at the Joint Genome Institute on either the Illumina HiSeq-2000 1TB or Illumina NovaSeq platforms (2x151bp reads) to generate metagenome-assembled genomes (MAGs). Read processing was performed in accordance with the JGI standard operating procedure 1064 using the jgi_meta_run.py (version 2.0.1) processing pipeline[92,93]. The resulting paired-end reads were then assembled using SPAdes assembler (version 3.11) using a range of Kmers with the parameters "spades.py -m 2000 –tmp-dir tmp.5774024.0 -o spades3 –only-assembler -k 33,55,77,99,127 –meta -t 32 –1 reads1.fasta –2 reads2.fasta"[94]. The entire filtered read set was mapped to the final assembly and coverage information was generated using bbmap (version 37.78) using default parameters except ambiguous=random (https://jgi.doe.gov/data-and-tools/bbtools/). Raw sequence data in FASTQ format are available on NCBI SRA and analysis and assemblies are available in the JGI GOLD database. Project and accession numbers are listed in Supplementary Data 3.

## Genome-resolved metagenomic analysis and network mapping

Binning of assembled contigs into MAGs was performed using Meta-BAT v2.12.1[95] implemented through the Genomes OnLine Database (GOLD) based on the established JGI workflow, and completion and contamination was assessed using CheckM v1.0.12[96]. Taxonomic assignment was performed using GTDB (release 86) with GTDB-tk v0.2.2[97]. MAG quality and taxonomic assignment were augmented with RNA detection including noncoding RNA (ncRNA), tRNAs, and rRNA (5S, 16S, 23S) genes using tRNAscan-SE v2.0.6 in "bacterial" and "archaeal" search modes and cmsearch from the INFERNAL v1.1.3 package against the Rfam 13.0 database using the trusted cutoffs parameter (–cut_tc)[98,99]. Reported hit overlapping by ≥1 bp and belonging to the same Rfam class were identified and the lower scoring of the two was removed.

High- and medium-quality MAGs were identified on the basis of current community standards (Supplementary Data 2)[100]. MAGs with >90% completeness, <5% contamination, and at least one of each rRNA subunit gene were retained as "high-quality" MAG populations. MAGs with >50% completeness, <10% contamination, and at least one rRNA subunit gene were retained as "medium quality" MAG populations. High- and medium-quality MAGs were uploaded as draft genomes to NCBI MAGs and were functionally annotated using the JGI's IMG workflow, which implements NCBI RPSBLAST to assign COG IDs to COG database v2014, EBI's pfam_scan tool (which uses HMMER v3.0) to assign PFAM IDs to Pfam database v30, LAST to assign KEGG KO Terms from IMG genes to the KEGG database v77.1, and LAST to assign EC numbers from IMG genes (using KO Terms) with a homology-based approach with a local installation of BLASTn v2.5.0[93].

DNA sequences of ASVs from the time series (~273 bp) were then searched as queries against a custom reference database of SSU rRNA gene sequences encoded in high- or medium-quality MAGs binned from the 17 metagenomes. Hits with >99.5% nucleotide identity over the full length of the query region were accepted (i.e., ASV V4 rRNA sequences with 0 or 1 mismatch along the 273 bp fragment to a MAG V4 rRNA region). For each of the 233 unique ASV-MAG pairs identified using this method, the GTDB taxonomy of each was compared and only pairs with a matching lowest-common-ancestor taxonomy were accepted (i.e., if an ASV was annotated to the order level and its MAG to the family level, the order level taxonomy must agree between the ASV and MAG). This resulted in 186 query-reference hits with coherent taxonomies, which were considered as ASV-MAG pairs and allowed for MAGs to be mapped onto ASV nodes in the co-occurrence network. Based on functional annotations from the JGI IMG workflow, the presence of methanogenic and potential syntrophic metabolic pathways in a select set of MAGs (i.e., those forming strong subnetworks in the co-occurrence analysis) were inferred from key encoded enzymes and electron carriers. Marker genes were selected to cover the necessary and sufficient steps of methanogenesis and syntrophy by known pathways[56,101–103].

## Metatranscriptomic sequencing and read mapping

Total RNA was extracted (from the same physical samples as for DNA) using the RNeasy Power Soil kit from Qiagen according to manufacturer specifications. RNA quality was checked using a bioanalyzer (Agilent) to measure concentration and RNA integrity number (RIN) a measure of the quality and integrity of the RNA. Samples were also measured on the NanoDrop (Thermo Fisher) for concentration and the 260/280 value to assess RNA purity. A total of 14 RNA extracts with 260/280 > 1.90 and RIN ≥ 5.0 were sent to GeneWiz for high-throughput sequencing of the community mRNA using "Standard RNA-seq" package. Details of this procedure are proprietary; more information can be found here (https://www.genewiz.com/Public/Services/Next-Generation-Sequencing/RNA-Seq/). Briefly, total RNA underwent rRNA depletion to concentrate mRNA transcripts, which were library prepped and sequenced Illumina HiSeq-2000 1TB platform (2 × 150bp reads). Raw reads were adapter trimmed, quality trimmed, and quality filtered using TrimmomaticSE (v0.35) with the parameters -phred33, LEADING:3 TRAILING:3 SLIDINGWINDOW:4:15 MINLEN:36. Quality-filtered RNA-seq read data in FASTQ format were uploaded to the NCBI under BioProject PRJNA902729 as BioSamples SAMN34106045 - SAMN34106058.

To infer and quantify gene expression of protein-coding genes on contigs assembled from each metagenome, quality-filtered transcripts from each sample were mapped to their respective metagenome library contigs using bowtie2 v2.2.5 with parameters –*end-to-end* and –*very-sensitive*[104]. The resulting BAM mapping files were further processed to identify and remove PCR duplicates using samtools and the MarkDuplicates function from Picard Toolkit (http://broadinstitute.github.io/picard/). The mRNA copy number of genes from each metagenome was then calculated using a protocol designed by the Environmental Genomics Group SciLifeLab at KTH Stockholm (https://metagenomics-workshop.readthedocs.io/en/latest/annotation/quantification.html). Briefly, HTSeq was used to count the number of reads mapped to features (e.g., genes, tRNA, etc.) on reference contigs using the GFF3 feature file generated during metagenome analysis. The number of metatranscript reads mapped to each feature were then normalized using the TPM (Transcripts per million) method. The resulting TPM tables for each of the 14 metagenome-metatranscriptome pairs (containing rows of gene features with TPM expression values) were parsed to calculate the percentage of metatranscript reads mapped to MAG bins and to associate TPM to functional pathways of interest in MAGs from the co-occurrence subnetworks (Supplementary Data 3).

## Reporting summary

Further information on research design is available in the Nature Portfolio Reporting Summary linked to this article.

# Data availability

Data generated for this study can be found deposited in public repositories. For 16S rRNA amplicon sequences, reference the NCBI GenBank database under BioProject PRJNA902729 as BioSamples SAMN32228586 - SAMN32228684. For whole-genome shotgun

metagenome sequences, reference the NCBI SRA database under BioProjects PRJNA502381, PRJNA502380, PRJNA518419, PRJNA502378, PRJNA502379, PRJNA502376, PRJNA502377, PRJNA502375, PRJNA620832, PRJNA620833, PRJNA620758, PRJNA620835, PRJNA620756, PRJNA620838, PRJNA620839, PRJNA620843, and PRJNA620754 (See Supplementary Data 3 for details and associations to JGI GOLD analysis accessions). For shotgun metatranscriptomic sequences, reference the NCBI SRA database under BioProject PRJNA902729 as BioSamples SAMN34106045 - SAMN34106058.

## Code availability
While all open source and freely available tools were used for this analysis, a markdown of the code used to conduct statistical analysis and create data visualizations in R v4.1.2 is referenced in the Supplementary Software.

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

## Acknowledgements

This work was performed under the auspices of the Natural Sciences and Engineering Research Council (NSERC) of Canada, the Canada Foundation for Innovation (CFI), and the US Department of Energy (DOE) Joint Genome Institute (JGI) and the Facilities Integrating Collaborations for User Science (FICUS) JGI-EMSL (Environmental Molecular Science Laboratory) project (50967) supported by the Office of Science of US DOE Contract DE-AC02-05CH11231. Essential automation support came from the Biofactorial high-throughput biology facility in the Life Sciences Institute at the University of British Columbia. We would like to thank Connor Morgan-Lang, Kateryna Ievdokymenko, Julia Anstett, and Siddarth Raghuvanshi for insightful conversations regarding data analysis, Michael Yoon for helping with sample processing, and Tom Pfeifer in the Biofactorial automation facility at UBC for help in establishing workflows and conducting extractions on the Access workstation. We thank Metro Vancouver staff in the Wastewater Treatment Operations and Collaborative Innovation groups for initiating and supporting this work to better understand microbial interactions in its AD systems.

## Author contributions

B.K., N.F., R.M., S.C., and S.J.H. designed experiments and sampling campaign. N.F. and S.C. were responsible for physicochemical measurements. B.K., R.M., A.N.N., and S.J.H. performed the microbiological data analyses. B.K. wrote analysis and figure generation code and performed integrative analysis for DNA and RNA data with input from S.J.H. M.K. contributed to figure design and interpretation. B.K. wrote the manuscript with input from N.F., R.M., A.N.N., S.C., M.K., and S.J.H.

## Competing interests

S.J.H. is a co-founder of Koonkie Inc., a bioinformatics consulting company that designs and provides scalable algorithmic and data analytics solutions in the cloud. The remaining authors declare no competing interests.
