## [Peer Review File · Nature Communications]

Genome-resolved correlation mapping links microbial community structure to metabolic interactions driving methane production from wastewaterEditorial Note: This manuscript has been previously reviewed at another journal that is not operating a transparent peer review scheme. This document only contains reviewer comments and rebuttal letters for versions considered at *Nature Communications*.

Reviewer #1 (Remarks to the Author):

Although I appreciate the authors efforts in revising the introduction, I unfortunately still do not agree with the authors claims.

Though I agree that, compared to previous studies, the authors sample at a higher resolution, implement a different correlation method, and sequence deeper, is there clear new insight gained? As pointed out by both reviewers, correlation between syntrophic and methanogenic populations is not new (*Smithella*, *Syntrophaceae*, *Syntrophomonas*, *Syntrophorhabdus*, *Christensenella*, *Bacteroidales*, and *Synergistetes* have all been reported to be capable of syntrophic interactions with methanogens). There are no new physiologies or interactions discovered/proposed here. The authors also point out themselves in the response letter, the conclusions are all putative and not necessarily novel.

Also, the problem statement presented here seems extremely vague to me. What were the knowledge gaps left by previous studies? The authors indicate in the response letter that:
(i) While we do now have a detailed foundational understanding of the biochemistry, taxa, and metabolic reactions associated with biomethane production (mostly thanks to the reductionist lab-scale approaches), we believe that there is now a need for data-rich time-resolved studies of AD operations at scale to help translate lab studies into results and to discover potentially new insights from ADs "in the wild".
(ii) most studies examining full-scale reactors do not have the time resolution, high-throughput molecular methods, association with detailed process data, or amount of sequence data we use in our work.

As the authors acknowledge themselves, many studies on anaerobic digestion target full-scale (see a short list of examples attached at the end). Although the authors suggest that the novelty is in the data and methods, if the data generated and methods applied are truly valuable, it should provide new valuable insight. As for the methods, the statistical analyses and metagenomics analyses use simple pre-existing packages/software and the high-throughput molecular methods use highly conventional Illumina-based amplicon and metagenomic sequencing. It's important to point out that, though the authors highlight their high sampling frequency, only one wastewater treatment plant is studied. The authors claim that the presented dataset is valuable to the research and industry community but how is this dataset translatable to other systems? For example, how can one generalize an interaction predicted from correlation observed in a single wastewater treatment plant to other treatment plants (especially given the complexity of anaerobic digester communities)? Many studies sample less frequently but study multiple wastewater treatment plants to get a more generalized view of the behavior of individual populations under different contexts.

Some full-scale studies (some may overlap with the list the authors provided in the response letter)

Jiang et al *Water Research* 2021 <https://doi.org/10.1016/j.watres.2021.116871>

Nobu et al *Microbiome* 2020 <https://doi.org/10.1186/s40168-020-00885-y>

Campanaro et al *Biotechnology for Biofuels* 2020 <https://doi.org/10.1186/s13068-020-01679-y>

Mei et al *Water Research* 2017 <https://doi.org/10.1016/j.watres.2017.07.050>

Reviewer #2 (Remarks to the Author):

The paper by Brandon Kieft and colleagues underwent a thorough examination to determine whether the raised criticisms had been addressed. This revised version of the manuscript effectively addresses and clarifies most of the weaknesses identified in the previous version, rendering it more robust overall. The revision brings about significant improvements; however, there are three specific points highlighted in the review that I recommend the Authors investigate further for better verification.

Previous comment - Line 184-185: It is unclear how Authors checked for DNA contaminations using Nanodrop. Please clarify.

Answer from the Authors: We were attempting to extract only RNA using the protocol. Values lower than 2.0 (pure RNA) from the Nanodrop's 260/280 indicated contamination with DNA or protein that was noted when submitting to the sequencing provider. No values below 1.9 were obtained and there was no indication of down stream effects from protein or DNA in the extracts. Line 234 present version. My comment here is that, according to my knowledge, it is very hard to identify DNA contamination using Nanodrop. For sure analysis using a fluorometer (e.g. Qbit Bioanalyzer) is much more reliable. I suggest the Authors to revise this sentence or to provide reference from literature regarding the procedure.

Previous comment -Lines 185-186: You should report the procedures used to generate RNA-seq libraries in order to facilitate comparisons with other datasets and to increase reproducibility.

Answer from the Authors: Thank you for asking for more detail here. We have reported the software used for transcript processing and upstream analysis, and added the SRA accessions for read data access for reproducibility and inclusion in other studies.

Line 237 present version. My comment here is that, I was not referring to data management but to the protocols used for libraries preparation for RNA sequencing. I suggest to check and add missing information (if needed).

Previous comment - Lines 411-435. Are these functional characteristics determined considering taxonomic information only? Did you consider the possibility to infer functional properties based on the gene content of strictly related species (e.g. a PICRUSt -like approach).

Answer from the Authors: No functional information in this work was inferred from taxonomic annotations alone or by using functional inference software against database representatives (e.g., PICRUSt). All functional information we describe (i.e., functions overlaid onto the co-occurrence network) is derived from binned MAGs that had a 16S rRNA V4 gene sequence that was homologous to an ASV 16S rRNA V4 sequence.

My suggestion here is to check the manuscript and clarify this aspect in order to clarify very well the procedure used.

REVIEWERS' COMMENTS

Reviewer #1 (Remarks to the Author):

Although I appreciate the authors efforts in revising the introduction, I unfortunately still do not agree with the authors claims.

Though I agree that, compared to previous studies, the authors sample at a higher resolution, implement a different correlation method, and sequence deeper, is there clear new insight gained? As pointed out by both reviewers, correlation between syntrophic and methanogenic populations is not new (Smithella, Syntrophaceae, Syntrophomonas, Syntrophorhabdus, Christensenella, Bacteroidales, and Synergistetes have all been reported to be capable of syntrophic interactions with methanogens). There are no new physiologies or interactions discovered/proposed here. The authors also point out themselves in the response letter, the conclusions are all putative and not necessarily novel.

Also, the problem statement presented here seems extremely vague to me. What were the knowledge gaps left by previous studies? The authors indicate in the response letter that:

(i) While we do now have a detailed foundational understanding of the biochemistry, taxa, and metabolic reactions associated with biomethane production (mostly thanks to the reductionist lab-scale approaches), we believe that there is now a need for data-rich time-resolved studies of AD operations at scale to help translate lab studies into results and to discover potentially new insights from ADs “in the wild”.

(ii) most studies examining full-scale reactors do not have the time resolution, high-throughput molecular methods, association with detailed process data, or amount of sequence data we use in our work.

As the authors acknowledge themselves, many studies on anaerobic digestion target full-scale (see a short list of examples attached at the end). Although the authors suggest that the novelty is in the data and methods, if the data generated and methods applied are truly valuable, it should provide new valuable insight. As for the methods, the statistical analyses and metagenomics analyses use simple pre-existing packages/software and the high-throughput molecular methods use highly conventional Illumina-based amplicon and metagenomic sequencing. It's important to point out that, though the authors highlight their high sampling frequency, only one wastewater treatment plant is studied. The authors claim that the presented dataset is valuable to the research and industry community but how is this dataset translatable to other systems? For example, how can one generalize an interaction predicted from correlation observed in a single wastewater treatment plant to other treatment plants (especially given the complexity of anaerobic digester communities)? Many studies sample less frequently but study multiple wastewater treatment plants to get a more generalized view of the behavior of individual populations under different contexts.

Some full-scale studies (some may overlap with the list the authors provided in the response letter):

Jiang et al Water Research 2021 <https://doi.org/10.1016/j.watres.2021.116871>

Nobu et al Microbiome 2020 <https://doi.org/10.1186/s40168-020-00885-y>

Campanaro et al Biotechnology for Biofuels 2020 <https://doi.org/10.1186/s13068-020-01679-y>

Mei et al Water Research 2017 <https://doi.org/10.1016/j.watres.2017.07.050>

We thank the reviewer for providing additional critical commentary on the novelty and relevance of this work, and for again highlighting prior efforts to work at municipal scales with an eye toward characterizing a core AD microbiome. Such a core has been considered an important foundation for developing more efficient process controls and monitoring tools motivating large-scale comparative studies such as those indicated by the reviewer. Indeed, the reference list provided are all papers cited in our work, which aimed to leverage information from these full-scale AD studies in the discussion. It should be noted that one common theme of large-scale studies integrating across multiple distinct AD systems is heterogeneity, making a single core microbiome difficult to define. Rather, communities tend to assemble based on operating conditions including feedstock, pre-treatment status, retention times, temperature, salinity, and pH. The paper by Mei et. al., provides a particularly good description of this heterogeneity across a global sampling of 51 municipal wastewater treatment plants.

Microbial community assembly is an important ecological concept along with conditionality, redundancy, degeneracy, and resilience to name a few. One could argue that discovering ecological design principles is the next frontier in developing more robust AD systems that are increasingly optimized for specific bioconversion or waste resource recovery functions. In this light, time-resolved studies of specific AD systems may in fact be more relevant to developing a productive ecological perspective than global comparisons for core taxa or conserved metabolic functions. Conducting such time-resolved studies is a non-trivial enterprise that requires long-term relationship building with plant operators and access to resources and metadata that can be extremely difficult to obtain outside of large consortia or foundation sponsored programs. There are in fact very few time-resolved studies of municipal wastewater treatment plants available in the public domain. We hope that this serves to make our overall problem statement less vague.

In the Canadian context our study is an outlier that required close collaboration with Metro Vancouver over a two-year period that included regular interactions with plant operators and project managers to facilitate sample collection, identify opportunities for changing process configuration for large-scale experimentation, and guide biological inquiries of interest to the user partner. Through these conversations we determined that the use of network theory to identify potential metabolic interactions in the AD milieu would be of interest and potential application to plant operators. Correlation networks are useful statistical constructs that can guide hypothesis development and testing with respect to groups of interacting microorganisms. Time-resolved correlation networks are uncommon in microbial ecology and genome-resolved

temporal networks with cognate transcriptional profiles are even less prevalent. AD systems present an interesting environmental context in which to conduct use-inspired research that also addresses more foundational aspects of microbial ecology.

The reviewer indicates that “There are no new physiologies or interactions discovered/proposed here. The authors also point out themselves in the response letter, the conclusions are all putative and not necessarily novel.” Metagenomic studies, are by their very nature hypothesis building exercises that can provide associations, correlations and predictions used to build various types of models. We are simply agreeing with the truth and tempering claims in a responsible manner. Our correlation-based approach offers an opportunity to infer known and novel interactions but needs some form of cross-validation to be ultimately useful. Finding known metabolic interactions in the network is certainly one way to build confidence in the method.

Genome-resolved correlation network mapping is not a standard method despite the reviewer’s sentiment regarding pre-existing software packages, and we are trying to develop its applicability in the AD milieu in a systematic manner that starts with conventional diversity profiling using amplicon sequence variants followed by mapping metagenome assembled genomes onto cognate network nodes to increase confidence in potential functional metabolic linkages between them. This approach is certainly more direct than transitive inference using tools like PICRUSt with the understanding that all open reading frame predictions are hypotheses. While it is true that we did not discover a completely novel form of symbiosis using this approach, we did identify a previously unrecognized linkage between a putative syntrophic acetate oxidizing microorganism and methanogen under a specific process configuration of interest to plant operators. This knowledge allows us to develop more targeted experiments including phylogenetic stains, cell sorting or stable isotope probing methods to further test and validate hypotheses generated from the correlation network. Again, we acknowledge that our current work is limited to in silico validation of a subset of interactions within the network and does not make full use of the multi-omic data generated for the complete time-series. However, when applying new methods at scale, it is important to take deliberate incremental steps. Subsequent stories remain under development that leverage these data in more comparative contexts, and we look forward to making the time-series data as accessible and useful to the user community as possible.

Reviewer #2 (Remarks to the Author):

The paper by Brandon Kieft and colleagues underwent a thorough examination to determine whether the raised criticisms had been addressed. This revised version of the manuscript effectively addresses and clarifies most of the weaknesses identified in the previous version, rendering it more robust overall. The revision brings about significant improvements; however, there are three specific points highlighted in the review that I recommend the Authors investigate further for better verification.

We thank the reviewer for their positive response to our revisions and try to address the remaining issues below.

Previous comment - Line 184-185: It is unclear how Authors checked for DNA contaminations using Nanodrop. Please clarify.

Answer from the Authors: We were attempting to extract only RNA using the protocol. Values lower than 2.0 (pure RNA) from the Nanodrop's 260/280 indicated contamination with DNA or protein that was noted when submitting to the sequencing provider. No values below 1.9 were obtained and there was no indication of downstream effects from protein or DNA in the extracts. Line 234 present version. My comment here is that, according to my knowledge, it is very hard to identify DNA contamination using Nanodrop. For sure analysis using a fluorometer (e.g. Qbit Bioanalyzer) is much more reliable. I suggest the Authors to revise this sentence or to provide reference from literature regarding the procedure.

We apologize for the confusion. This sentence was revised to focus specifically on the A260/A280 ratio and not make any reference to "contamination". The reason for measuring this spectrophotometric value is that sequencing centers often require it to be reported because they have a specific nucleic acid QC range that they will accept for a given sequencing service e.g., pure RNA has an A260/A280 ration of 2.0 and pure DNA has an A260/A280 ration of 1.8).

Previous comment -Lines 185-186: You should report the procedures used to generate RNA-seq libraries in order to facilitate comparisons with other datasets and to increase reproducibility. Answer from the Authors: Thank you for asking for more detail here. We have reported the software used for transcript processing and upstream analysis and added the SRA accessions for read data access for reproducibility and inclusion in other studies.

Line 237 present version. My comment here is that, I was not referring to data management but to the protocols used for libraries preparation for RNA sequencing. I suggest to check and add missing information (if needed).

Thank you for this comment. We recognize that transparency and access are essential aspects of reproducible science. We attempted to get specific information about the rRNA depletion and library preparation procedures used by the sequencing service provider (GeneWiz, now Azenta). Although they would not provide specific details claiming proprietary know-how, we amended the methods section to identify the service provider, and provide basic steps taken with a link to their product page that contains additional information relevant to their commercial RNA-Seq pipeline.

Previous comment - Lines 411-435. Are these functional characteristics determined considering taxonomic information only? Did you consider the possibility to infer functional properties based on the gene content of strictly related species (e.g. a PICRUSt -like approach). Answer from the Authors: No functional information in this work was inferred from taxonomic annotations alone or by using functional inference software against database representatives (e.g., PICRUSt). All functional information we describe (i.e., functions overlaid onto the co-occurrence network) is derived from binned MAGs that had a 16S rRNA V4 gene sequence that

was homologous to an ASV 16S rRNA V4 sequence. My suggestion here is to check the manuscript and clarify this aspect in order to clarify very well the procedure used.

Thanks for pointing out the ambiguities here. We have gone through the manuscript and changed some sentences that referenced the functional inference of network nodes to clarify that function/metabolic data came directly from the paired metagenomes and not from database inferences. While we recognize the utility of inference tools like PICRUSt in the absence of metagenomic data when an appropriate reference genome data database exists, our study explicitly wanted to make more direct inferences by leveraging the mapping of amplicon sequence variants to cognate metagenome assembled genomes in the correlation network.